# Direction-of-Arrival Estimation via Sparse Bayesian Learning Exploiting Hierarchical Priors with Low Complexity

**DOI:** 10.3390/s24072336

**Published:** 2024-04-06

**Authors:** Ninghui Li, Xiaokuan Zhang, Fan Lv, Binfeng Zong

**Affiliations:** 1Graduate School, Air Force Engineering University, Xi’an 710051, China; li6667778882022@163.com (N.L.); lf995407681@163.com (F.L.); 2Air Defence and Antimissile School, Air Force Engineering University, Xi’an 710051, China; ezxk@sina.com

**Keywords:** direction-of-arrival estimation, sparse Bayesian learning, hierarchical priors, block sparse

## Abstract

For direction-of-arrival (DOA) estimation problems in a sparse domain, sparse Bayesian learning (SBL) is highly favored by researchers owing to its excellent estimation performance. However, traditional SBL-based methods always assign Gaussian priors to parameters to be solved, leading to moderate sparse signal recovery (SSR) effects. The reason is Gaussian priors play a similar role to $l_{2}$ regularization in sparsity constraint. Therefore, numerous methods are developed by adopting hierarchical priors that are used to perform better than Gaussian priors. However, these methods are in straitened circumstances when multiple measurement vector (MMV) data are adopted. On this basis, a block-sparse SBL method (named BSBL) is developed to handle DOA estimation problems in MMV models. The novelty of BSBL is the combination of hierarchical priors and block-sparse model originating from MMV data. Therefore, on the one hand, BSBL transfers the MMV model to a block-sparse model by vectorization so that Bayesian learning is directly performed, regardless of the prior independent assumption of different measurement vectors and the inconvenience caused by the solution of matrix form. On the other hand, BSBL inherited the advantage of hierarchical priors for better SSR ability. Despite the benefit, BSBL still has the disadvantage of relatively large computation complexity caused by high dimensional matrix operations. In view of this, two operations are implemented for low complexity. One is reducing the matrix dimension of BSBL by approximation, generating a method named BSBL-APPR, and the other is embedding the generalized approximate message passing (GAMB) technique into BSBL so as to decompose matrix operations into vector or scale operations, named BSBL-GAMP. Moreover, BSBL is able to suppress temporal correlation and handle wideband sources easily. Extensive simulation results are presented to prove the superiority of BSBL over other state-of-the-art algorithms.

## 1. Introduction

DOA estimation has advanced obviously due to several technique leaps in the last four decades, and the attained achievements are widely applied to communications, radar, sonar, and navigation. To be specific in communication [1,2], DOA estimation is essential in channel estimation, wireless communications, microphone localization, vehicular communications [3], Reconfigurable Intelligent Surfaces (RIS), and corresponding research focuses, including the RIS-based vehicle DOA estimation method [4,5,6].

Among the technique leaps in DOA estimation, compressed sensing (CS) and sparse recovery (SR) have played important roles in the last decade [7,8]. Compared with traditional algorithms based on beamforming or subspace techniques [9,10,11,12,13], sparsity-based estimators have achieved technique leaps since CS and SR can mitigate the requirements for high signal-to-noise ratios (SNRs) and abundant snapshots [14]. Moreover, it has been proven that sparsity-based estimators have remarkable advantages, such as good robustness to correlation and high estimation accuracy.

In recent years, relevant studies on DOA estimation mainly focus on sparse array configuration [15,16,17] or sparsity-based method innovation [18,19]. For the latter, three categories can be summarized out of massive documents. (i) The first is based on norm optimization [20,21,22,23,24,25,26,27,28], (ii) the second is exploiting basis pursuit [29,30], and (iii), and the third is utilizing sparse Bayesian learning (SBL) [31,32,33,34,35,36,37,38,39,40,41,42,43,44,45,46,47]. In particular, SBL-based methods are popular and highly favored owing to incomparable comprehensive performance beyond both norm optimization and basis pursuit [48]. 

From a Bayesian perspective, sparse Bayesian learning is a probabilistic method that achieves $l_{p}$-norm minimization by assigning sparse priors to the signal of interests for sparse signal recovery (SSR). In the single measurement vector (SMV) model, SBL can retain a desirable property of the $l_{0}$-norm diversity measure, i.e., the global minimum is achieved at the maximally sparse solution [48] and produced a more limited constellation of local minima. In practice, the most used is the MMV model, so MSBL is developed [49]. Theoretically, in [49], the adopted empirical Bayesian prior plays an important role in estimating a convenient posterior distribution over candidate basis vectors based on the concept of automatic relevance determination. That implies that the used priors assigned to signals of interest can enforce a common sparsity profile and consistently place the prominent posterior mass on the appropriate region for sparse recovery. In other words, the adopted priors dominate the sparsity performance of SBL and indicate the ability to carry out $l_{p}$-norm minimization. Despite this, it is still unclear which prior is the best for sparse recovery, but a hierarchical Bayesian framework representing Laplace priors has been proven to be prominent in [50]. Therefore, hierarchical priors are widely attractive, and many corresponding works are developed and presented. For instance, hierarchical synthesis lasso (HSL) priors for representing the same small subset of features are created for enforcing a proper sparsity profile of signal vectors [46], and hierarchical priors are adopted to consider unknown mutual coupling and off-grid errors. No matter how hierarchical priors are used, SBL has to face the difficulty caused by the MMV model. To be specific, SBL needs prior assumption, i.e., the uncorrelation between different measurement vectors, while the assumption may not be satisfied in practice. Moreover, the solution of matrix form is not always easy to handle and is even prohibitive due to the large complexity during the learning process. On this basis, vectorizing the MMV model seems to be an optimal selection since the solution can be transformed into the vector form. A successful realization is developed in [32], and the used block-sparse model indeed contributes much to the whole designed algorithm, although the main focus is the real-valued transformation. Regretfully, that work adopts Gaussian priors rather than hierarchical priors, so its sparsity performance is bound to be limited, no matter how impressive and excellent the running efficiency is. Recently, there have been a few documents that show the combination of SBL and deep learning (DL) [51,52], attracting widespread attention among numerous researchers. With big data and artificial intelligence, DL gradually arises and applies to image processing, signal processing, classification, recognition, etc. The obvious advantages of DL are its adaptability to complicated practical cases and its high running efficiency. In signal processing, many researchers use derived iterative processes to design corresponding neural networks so that the proposed algorithms operate with low complexity burdens, and the designed networks are named deep unrolled neural networks. For example, an SBL-based algorithm is unfolded into a layer-wise structure with a set of introduced trainable parameters in [51], which is beneficial for channel estimation. In addition, a model-driven DL detector is developed based on variational Bayesian inference in [52]. Based on deep unrolled networks, the detector is able to capture channel features that may be important but neglected by model-based methods, including SBL-based methods. Although the algorithms based on both SBL and DL are not applied to DOA estimation, the disadvantages lie in the inexplicability caused by data-driven DL techniques. In fact, the lack of theoretical guarantee is always constraining the development and application of DL. Collectively, SBL-based methods face a conflict between prominent sparsity performance achieved by hierarchical priors with low computational complexity. Methods based on both SBL and DL are pending further development, especially in theoretical explicability. 

In this paper, we creatively solve the contradiction between sparsity performance with complexity burdens in SBL-based methods. On the one hand, hierarchical priors are still adopted to indirectly enhance sparsity. On the other hand, the increased complexity essentially caused by the hierarchical priors in MMV models is tactfully reduced by two operations based on the transformed block-sparse model. In fact, the balance of both sparsity and complexity is the main innovation point in this paper. Not only that, but the block-sparse model provides great convenience for complexity reduction by decreasing matrix dimensions in terms of matrix operation properties, and hierarchical priors create an opportunity to embed the generalized approximate message passing (GAMB) technique for solving marginal distributions so as to reduce complexity greatly. 

To be specific, as the thick and inevitable MMV model restricts the SBL to some extent, we directly vectorize the MMV model, resulting in a block-sparse model that is convenient to carry out Bayesian learning. Unluckily, the vectorization expands the model dimensions, leading to large complexity in later Bayesian learning. Despite this, block-sparse Bayesian learning, named BSBL, is still analytically derived and developed. Moreover, it is worth noting that the complexity of BSBL is reluctantly acceptable owing to our reasonable design of the iterative process. In order to further achieve complexity reduction, two operations are introduced. One is based on matrix operation properties to decrease the matrix dimensions. Specifically, we analyze the complexity of each iterative formula and select the one containing operations of large computation burdens. Later, the selected formulas of BSBL are simplified and approximated by terms of operational properties of Kronecker products and some reasonable preconditions, so the faster version is named BSBL-APPR. The other is based on the famous generalized approximate message passing (GAMB) technique. GAMP is developed to solve approximate marginal posteriors, which are exactly applicable in BSBL since BSBL is derived by iterative hyperparameters originating from marginal distributions. Therefore, the GAMP technique is able to be embedded into BSBL, and the only additional work is to derive the iterative process of GAMP so that GAMP is useful in our block-sparse model. Moreover, GAMP is able to decompose the high dimensional matrix operations into vector or scale operations, which achieves complexity reduction well. BSBL with embedded GAMP is named BSBL-GAMP. In addition, many SBL-based methods consider little about the intractable wideband cases. Since wideband sources are able to be regarded as a superposition of many narrowband sources, we extend the proposed BSBL to wideband cases. The whole algorithm for wideband cases is derived and finished. Last but not least, the temporal correlation, which is not often considered in SBL, is modeled in the block-sparse model. Therefore, all the above methods are able to suppress temporal correlation.

In summary, the contributions of this paper are as follows:
Hierarchical priors are adopted to enhance sparsity, and a block-sparse model is generated to carry out Bayesian learning easily. Hierarchical priors play an important role in $l_{p}$-norm optimization and outperform Gaussian priors in sparsity constraint, indirectly resulting in better sparsity performance. In the MMV case, the equivalently transformed block-sparse model laid a foundation for complexity reduction. Combining hierarchical priors with the block-sparse model allows for the balance of sparsity and complexity;Two operations are created to reduce complexity based on the block-sparse model. One exploits matrix operation properties to approximate high-dimension operations of derived formulas in the iterative process, while the other leverages the GAMP technique to simplify the iteration for computing marginal distributions, so that the matrix operations are decomposed into vector or scale operations;For wideband sources appearing in practice, the proposed BSBL is extended to be applicable in terms of the decomposition of wideband signals into narrowband ones. Moreover, the temporal correlation is considered by introducing a temporally correlated matrix into our data model. The designed iterative process of BSBL is able to be robust to temporal correlation.

The rest of this paper is organized as follows. In Section 2, the DOA estimation problem is abstracted from radar detection. Furthermore, the DOA estimation is equivalently transformed into an SSR problem based on the exploited block-sparse model. In Section 3, the traditional SBL based on our model is briefly introduced and derived, and, its defects are presented by simple analysis. In Section 4, the proposed BSBL, BSBL-APPR, BSBL-GAMP, and BSBL for wideband cases are analytically derived, and corresponding iterative processes are presented. In Section 5, the performance of the proposed methods is evaluated comprehensively. In Section 6, conclusions are drawn. 

For the sake of convenience, the notations are listed in Table 1.

## 2. Problem Reformulation

In radar or sonar detection, for communication, localization, and navigation, the antenna array receiver can receive different signals from various directions. Taking far-field radar detection as an example, adjacent antenna sensors have the same phase difference due to the plane electromagnetic wave if every adjacent couple has the same distance. Thus, a steer vector can be abstracted as a(θ)=[1,exp(−j2πdsin(θ)/λ),…,exp(−j2π(N−1)dsin(θ)/λ)]T∈ℂN×1. θ is the direction of the source signal. λ is the wavelength. d is the distance between adjacent sensors. N is the number of sensors. For all the K sources from different directions {θk=1K}, a steer matrix is yielded as A=[a(θ1),…,a(θK)]∈ℂN×K. A is vital in DOA estimation because it implies the directions of all the sources based on the whole antenna array composed of N sensors. Radio frequency signals were received by an antenna array, whose sensors transfer individually received signals to independent channels, in which down conversion is conducted to produce intermediate-frequency signals, i.e., baseband signals. Later, using prior signals to execute matched filtering at time l, the receiver can gain a complex reflection factor of all the K sources, i.e., s(l)=[s1,…,sk,…,sK]T∈ℂK×1. sk and ∀k=1,…,K are the products of complex reflection coefficients and Doppler shifts. Generally, radar takes pulse accumulation to enhance signal processing ability when the echo pulses are in the same coherent processing interval (CPI), in which little fluctuation happens between different originally received signals and processed signals. In other words, sources are motionless, and the processing signals are nearly consistent during a CPI. Last but not least, the noises of N channels in a CPI are indispensable and assumed to be mutually independent. Overall, the above entire process is briefly shown in Figure 1, where the signal model is expressed as
(1)X=AS+N.
where X=[x1,…,xL]∈ℂN×L is the ideal data of N channels in a CPI containing L snapshots. S=[s(1),…,s(l),…,s(L)]∈ℂK×L is the complex reflection factor matrix of the *K* sources in a CPI. N=[n1,…,nL] is the assumed noise matrix with each entry nl obeying an i.i.d. Gaussian distribution denoted as CN(0,α−1).

Let {θ¯m=1M} be M-grid sampling that covers spatial range [−π/2,π/2]. If sources are located on the grid exactly, (1) can be transformed into a sparse model expressed as
(2)X=A¯P+N.
where A¯=[a(θ¯1),…,a(θ¯M)]∈ℂN×M is the sparsely extended manifold matrix with the angle set {θ¯m=1M} generated by the M-grid sampling. As K≪M holds in most cases, P∈ℂM×L is the zero-padding version of S, with each row representing a potential source that maps the M-grid spatial angular sampling. 

Patently, processed data in a CPI are equivalently transformed into (2), which are typically sparse data in the MMV model. Now, the objective is to solute the sparse P with known data X and steer matrix $\bar{A}$. It is worth emphasizing two mainstream methods to handle (2). One is dealing with (2) directly, which is more difficult but has less computational complexity. The other is solving the vectorization (2), which is simple and intuitive but has large complexity caused by the vectorization. In this paper, the latter is selected because we are able to reduce the complexity tactfully. Without loss of generality, (2) is vectorized as
(3)x=Φp+n.
where x=vec(XT)∈ℂNL×1, Φ=A¯⊗IL∈ℂNL×ML, p=vec(PT)∈ℂML×1, and n=vec(NT)∈ℂNL×1. After the vectorization, a block-sparse vector p is yielded since the original P contains many zero rows. In addition, the matrix dimensions grow after the vectorization, leading to large complexity, but the problem will be solved by our approximation eventually. Based on the block-sparse model shown in (3), the DOA estimation is transformed into a sparse recovery problem, i.e., to solve p with known x and Φ.

## 3. Canonical SBL Method

To solve p in (3), the canonical SBL method is introduced and briefly shown as follows. According to (3), the likelihood is
(4)$$p(x|p;\alpha )\sim CN(\Phi p,\alpha^{-1}I_{NL}).$$

The prior of **p** is supposed to be
(5)$$p(p;\gamma_{i},\forall i=1,\ldots ,M)\sim CN(0,\Sigma_{0}).$$
where $\gamma_{i}$ is a hyperparameter representing a potential source, and $\Sigma_{0}$ is expressed as
(6)$$\Sigma_{0}=\left[\begin{array}{ccc} \gamma_{1}I_{L} & & \\ & \ddots & \\ & & \gamma_{M}I_{L} \end{array}\right].$$

Based on the Bayesian formula, the posterior of **p** is
(7)$$p(p|x;\alpha ,\gamma_{i},\forall i)=\frac{p(x|p;\alpha )p(p;\gamma_{i},\forall i)}{\int p(x|p;\alpha )p(p;\gamma_{i},\forall i)dp}.$$

The posterior of (7) is rigorously solved as a Gaussian distribution with mean and covariance as follows:(8)$$\mu_{p0}=\alpha \Sigma_{p0}\Phi^{H}x.$$
(9)$$\Sigma_{p0}=\Sigma_{0}-\Sigma_{0}\Phi^{H}{\left(\alpha^{-1}I+\Phi \Sigma_{0}\Phi^{H}\right)}^{-1}\Phi \Sigma_{0}.$$

The likelihood, prior, and posterior are uniquely determined by the hyperparameter set $\Theta =\{\gamma_{i},\alpha ,\forall i\}$. According to the maximum a posterior (MAP) criterion, the Expectation–Maximization (EM) algorithm [53] is used to maximize p(p|x;Θ). Here, p is treated as a hidden variable to obtain the relationship between the hyperparameter set (i.e., $\Theta^{new}$) and the old one (i.e., $\Theta^{old}$) by maximizing the following term.
(10)$$Q(\Theta^{new})=E_{p|x,\Theta^{old}}[ln\;p(p|x;\Theta )].$$

Omitting the specific derivation, the final iteration solutions of the hyperparameters are expressed as
(11)$$\gamma_{i}^{new}=\frac{tr[\Sigma_{p0}^{i}+\mu_{p0}^{i}{\left(\mu_{p0}^{i}\right)}^{H}]}{L},\forall i.$$
(12)$$\alpha^{new}=\frac{NL}{{\left\|x-\Phi \mu_{p0}\right\|}_{2}^{2}+tr(\Sigma_{p0}\Phi^{H}\Phi )}.$$
where $\Sigma_{p0}^{i}=\Sigma_{p0}([(i-1)L+1:iL],[(i-1)L+1:iL])$ and $\mu_{p0}^{i}=\mu_{p0}([(i-1)L+1:iL])$. 

The canonical SBL algorithm uses (8), (9), (11), and (12) iteratively to estimate γ until convergence, and the final $\gamma ={\left[\gamma_{1},\ldots ,\gamma_{M}\right]}^{T}$ is regarded as the solved p. However, recalling the whole process, two limitations are present: (i) the single Gaussian priors cannot enhance sparsity well and (ii) the computational complexity $O(M^{3}L^{3})$, dominated by (12), is usually unacceptable in practice. Consequently, we develop an SBL-based method to resolve (3) in this paper.

## 4. Proposed Methods

### 4.1. BSBL

Without loss of generality, the SBL-based methods are required to construct a Bayesian framework and then complete the corresponding Bayesian inference to develop an iterative algorithm.

#### 4.1.1. Bayesian Framework

Bayesian framework is composed of prior distributions of observed data and unknown variables. 

**Remark** **1.**
*According to the MAP criterion, priors are essential for SBL-based methods because the iterative process to be constructed is based on the derivatives of different variables so as to ensure the maximal posterior. Therefore, the prior distributions need to be clarified first.*


In this paper, the prior distribution (i.e., likelihood) of the observed data x is similar to (4), i.e.,
(13)$$p(x|p;\alpha ,\gamma )\sim CN(\Phi p,\alpha^{-1}I_{NL}).$$

For better sparsity performance, we propose hierarchical priors containing Gaussian and Gamma priors. The reason for selecting Gamma priors is that the Gamma distribution is the conjugate prior to the inverse variance of the Gaussian distribution [54]. As usual, the prior p obeys the i.i.d. complex Gaussian distribution, i.e.,
(14)p(p|γ)~CN(0,Σ).
where γ=[γ1,…,γM]T, γ−1=[γ1−1,…,γM−1]T, and Σ=diag(γ−1)⊗B. Note that B∈ℂL×L representing temporally correlated level is not equal to IL in the canonical SBL method. Therefore, the proposed method will be able to suppress temporal correlation, which will be tested and verified in Section 5. B is generally modeled as a Toeplitz matrix, i.e.,
(15)B=(1β⋯βL−1β1⋯βL−2⋮⋮⋱⋮βL−1βL−2⋯1).
where β is the complex correlation coefficient with the amplitude |β|∈[0,1] and phase arg(β)∈[0,2π].

γ obeys a Gamma distribution, i.e.,
(16)$$p(\gamma ;a,b)=\prod_{m=1}^{M}\frac{b^{a}}{\Gamma (a)}\gamma_{m}^{a-1}e^{-b\gamma_{m}}.$$
where $\Gamma (a)=\int_{0}^{\infty }x^{a-1}\exp(-x)dx$. a and b are the shape parameter and scale parameter. Then, a Gamma prior is applied to α so that
(17)$$p(\alpha ;c,d)=\frac{c^{d}}{\Gamma (c)}\alpha^{c-1}e^{-d\alpha }.$$
where c and d are the corresponding shape and scale parameters, respectively. 

It is worth emphasizing that the true prior distribution of p can be solved with Student’s t-distribution, which promotes sparsity better than the traditional Gaussian distribution [47].

#### 4.1.2. Bayesian Inference

Bayesian inference is necessary for the eventual iterative algorithm, and the crux is to deduce the posterior. Unfortunately, according to our Bayesian framework, the posterior is intractable. However, we just need to maximize the posterior by maximizing the evidence procedure, regardless of the analytical closed form of the posterior. Coincidentally, OGSBI [54] provides an example to maximize evidence, i.e., the marginal probability of the observed data x
(18)p(x)=∫∫∫p(x|p;α,γ)p(p|γ)p(γ;a,b)p(α;c,d)dpdγdα.

But, (18) is still intractable, so maximizing evidence seems to help little. However, variational inference is able to achieve it. It is necessary to explain variational inference before later derivation. Variational inference defines a function as a mapping that takes a function as the input and returns the value of the function as the output [55]. In the entropy field, the function is a probability distribution. When variational inference is applied to (18), the parameter vector (i.e., unknown stochastic variables) no longer appears because the parameters are absorbed into new probability distributions. Thus, (18) can be converted to an addressable form.

To be specific, we adopt variational Bayesian inference (VBI) [56] to address (18) by introducing a distribution q(Θ), where Θ={p,γ,α} is the parameter set of unknown variables. The introduced q(Θ) can simplify (18) and allow the logarithmic form of (18) to be divided into two parts, i.e.,
(19)$$\ln p(x)=\underset{F(q,\Theta )}{\underset{︸}{\int q(\Theta )\ln\frac{p(x,\Theta )}{q(\Theta )}d\Theta }}\underset{KL(q||p)}{\underset{︸}{-\int q(\Theta )\ln\frac{p(\Theta |x)}{q(\Theta )}d\Theta }}.$$
where p(x,Θ)=p(x|p;α,γ)p(p|γ)p(γ;a,b)p(α;c,d) is the product of priors and p(Θ|x) is the posterior. The specific derivation from (18) to (19) is complicated [55], so it is omitted here. F(q,Θ) is the lower bound of lnp(x) because KL(q||p)≥0 is the Kullback–Leibler divergence between q(Θ) and the posterior p(Θ|x). 

The significance of (19) is transforming the intractable lnp(x) into an approximated tractable F(q,Θ), so that maximizing lnp(x) is approximately equal to maximizing F(q,Θ). The lower bound F(q,Θ) is a functional in terms of q(Θ). In other words, F(q,Θ) is a mapping that takes as input a function q(Θ) and returns the value of the function as the output. Similar to the function derivative, maximizing F(q,Θ) requires some optimization over specific forms of q(Θ). In Bayesian inference, the commonly used form is factorization [56]. 

As performed in [56], to achieve the maximized F(q,Θ), q(Θ) is factorized into independent parts, i.e.,
(20)$$q(\Theta )=\prod_{i}q_{i}(\Theta_{i})=q(p)q(\gamma )q(\alpha ).$$
where q(p), q(γ), and q(α) correspond marginal distributions of the hidden variables p, γ, and α. $q_{i}(\Theta_{i})$ is expressed as
(21)$$q_{i}(\Theta_{i})=\frac{\exp{\langle \ln p(x,\Theta )\rangle }_{j\neq i}}{\int \exp{\langle \ln p(x,\Theta )\rangle }_{j\neq i}d\Theta_{i}}.$$

(21) is hard to compute, but its logarithmic form is easy to obtain, i.e.,
(22)$$\ln q_{i}(\Theta_{i})={\langle \ln p(x,\Theta )\rangle }_{j\neq i}+const.$$

Using (22), marginal distributions q(p), q(γ), and q(α) can be solved. 

**Step 1**: lnq(p) satisfies
(23)$$\ln q(p)={\langle \ln p(x|p,\alpha )p(p;\gamma )\rangle }_{q(\gamma )q(\alpha )}+const.$$

Utilizing (13), (14), and (23), q(p) can be solved as a Gaussian distribution, with mean and variance given by
(24)$$\mu_{p}=\langle \alpha \rangle \Sigma_{p}\Phi^{H}x.$$
(25)$$\Sigma_{p}={\left(\langle \alpha \rangle \Phi^{H}\Phi +{\langle \Sigma \rangle }^{-1}\right)}^{-1}.$$

Please refer to Appendix A for the proof. In general, (24) and (25) are equivalently transformed into beingless complex according to the properties of the matrix in [46].
(26)$$\mu_{p}=\langle \Sigma \rangle \Phi^{H}{\left({\langle \alpha \rangle }^{-1}I_{NL}+\Phi \langle \Sigma \rangle \Phi^{H}\right)}^{-1}x.$$
(27)$$\Sigma_{p}=\langle \Sigma \rangle -\langle \Sigma \rangle \Phi^{H}{\left({\langle \alpha \rangle }^{-1}I_{NL}+\Phi \langle \Sigma \rangle \Phi^{H}\right)}^{-1}\Phi \langle \Sigma \rangle .$$

**Step 2**: lnq(γ) satisfies
(28)$$\ln q(\gamma )={\langle \ln p(p;\gamma )p(\gamma ;a,b)\rangle }_{q(p)q(\alpha )}+const.$$

Using (14), (16), and (28), q(γ) is identified as a Gamma distribution, whose shape parameter, the m−th scale parameters, and the m−th element of the mean are as follows:(29)a¯=a+12.
(30)b¯m=b+12〈pmHB−1pm〉.
(31)〈γm〉=a¯b¯m.
where pm∈ℂL×1 is the m−th entry of p=[p1T,…,pMT]T and m=1,2,…,M. Please refer to Appendix B for the proof. (30) can be simplified further, i.e.,
(32)b¯m=b+12‖(B−1+1L×L−IL)⊙(μpmμpmH)‖1,1+12diag(B−1⊙Σpm).
where μpm is the m−th entry of μp=[μp1T,…,μpMT]T, Σpm=〈γm−1〉B, and m=1,2,…,M. Please refer to Appendix C for details of the derivation.

**Step 3**: lnq(α) satisfies
(33)$$\ln q(\alpha )={\langle \ln p(x|p,\alpha )p(\alpha ;c,d)\rangle }_{q(p)q(\gamma )}+const.$$

Using (13), (17), and (33), q(α) is also solved as a Gamma distribution with shape parameter $\bar{c}$, scale parameter $\bar{d}$, and mean 〈α〉 as follows: (34)$$\bar{c}=c+\frac{NL}{2}.$$
(35)$$\bar{d}=d+\frac{1}{2}\langle {\left(x-\Phi p\right)}^{H}(x-\Phi p)\rangle =d+\frac{1}{2}({\left\|x-\Phi \mu_{p}\right\|}_{2}^{2}+tr(\Sigma_{p}\Phi^{H}\Phi )).$$
(36)$$\langle \alpha \rangle =\frac{\bar{c}}{\bar{d}}.$$

Please refer to Appendix D for the proof. For the convenience of iteration, (35) is further simplified as
(37)$$\bar{d}=d+\frac{1}{2}({\left\|x-\Phi \mu_{p}\right\|}_{2}^{2}+tr(\Sigma_{p}\Phi^{H}\Phi ))=d+\frac{1}{2}{\left\|x-\Phi \mu_{p}\right\|}_{2}^{2}+\frac{1}{2}{\langle \alpha \rangle }^{-1}[ML-tr(\Sigma_{p}{\langle \Sigma \rangle }^{-1})].$$

Overall, q(p), q(γ), and q(α) are solved, and, alternately, are updating (26), (27), (31), (36) until some convergence criterion is satisfied. 

**Remark** **2.***According to [57], maximizing the lower bound* F(q,Θ) *guarantees convergence of the iterative optimization since each iteration leads to a nondecreasing value of* F(q,Θ)*. Therefore, the proposed method must converge at some point.*

#### 4.1.3. Off-Grid Correction

Recalling (2), P with angle set $\left\{{\bar{\theta }}_{m=1}^{M}\right\}$ is yielded by spatial discretization, causing estimation errors inevitably if sources are off grid. On this basis, the array steering vector of the i−th source is Taylor expanded around the nearest sampling grid denoted as ${\bar{\theta }}_{m_{i}}$, i.e.,
(38)$$a(\theta_{i})\approx a({\bar{\theta }}_{m_{i}})+a_{1}({\bar{\theta }}_{m_{i}})\Delta \theta_{i}.$$
where $\Delta \theta_{i}=\theta_{i}-{\bar{\theta }}_{m_{i}}\in [-\frac{1}{2}\left\|{\bar{\theta }}_{m}-{\bar{\theta }}_{m+1}\right\|,\frac{1}{2}\left\|{\bar{\theta }}_{m}-{\bar{\theta }}_{m+1}\right\|]$ and $a_{1}({\bar{\theta }}_{m_{i}})=a({\bar{\theta }}_{m_{i}}{)}^{\prime }$. Patently, the final objective is to solve $\Delta \theta_{i}$. Following the above similar principle based on Taylor expansion, Φ can be extended as
(39)$$\Phi (\lambda )=\Phi +\Phi^{\prime }diag(\lambda ).$$
where $\lambda ={\left[\lambda_{1},\ldots ,\lambda_{ML}\right]}^{T}$ with the i−th element is $\lambda_{i}\in [-\frac{1}{2}\left\|{\bar{\theta }}_{m}-{\bar{\theta }}_{m+1}\right\|,\frac{1}{2}\left\|{\bar{\theta }}_{m}-{\bar{\theta }}_{m+1}\right\|]$ and ∀i=1,…,ML. To solve λ, (3) can be used, i.e.,
(40)$$E(x-\Phi (\lambda )p)=x-\Phi (\lambda )\mu_{p}=0.$$

Combining (40) with (39), the following equation holds.
(41)$$\lambda ⊙\mu_{p}={\left[{\left(\Phi^{\prime }\right)}^{H}\Phi^{\prime }\right]}^{-1}{\left(\Phi^{\prime }\right)}^{H}(x-\Phi \mu_{p}).$$

Using (41), λ is solved and is still block sparse; thus, we squeeze it as $\lambda^{squ}={\left[\lambda_{1}^{squ},\ldots ,\lambda_{M}^{squ}\right]}^{T}$, where the m−th element $\lambda_{m}^{squ}=sum\{\lambda ([(m-1)L+1:mL])\}/L$. Here, if $\lambda_{m}^{squ}$ exceeds the interval $\left[-\frac{1}{2}\left\|{\bar{\theta }}_{m}-{\bar{\theta }}_{m+1}\right\|,\frac{1}{2}\left\|{\bar{\theta }}_{m}-{\bar{\theta }}_{m+1}\right\|\right]$, $\pm \frac{1}{2}\left\|{\bar{\theta }}_{m}-{\bar{\theta }}_{m+1}\right\|$ will be assigned to it. Letting the preliminary estimated DOA value vector be $\theta^{est1}={\left[{\hat{\theta }}_{1},\ldots ,{\hat{\theta }}_{k},\ldots ,{\hat{\theta }}_{K}\right]}^{T}$, the final DOA vector, denoted as $\theta^{est2}$, is solved as follows:(42)$$\theta^{est2}=\theta^{est1}+\lambda^{squ,est}.$$
where $\lambda^{squ,est}={\left[\lambda_{m_{1}}^{squ},\ldots ,\lambda_{m_{k}}^{squ},\ldots ,\lambda_{m_{K}}^{squ}\right]}^{T}$, $\lambda_{m_{k}}^{squ}$ is the corresponding angle compensation for ${\hat{\theta }}_{k}$. Using (42), grid errors are eliminated.

Overall, the whole BSBL algorithm is completed and summarized in Algorithm 1.
**Algorithm 1.** The proposed BSBL algorithm.**Initialization** (i) set the first iterative number k=0, $p^{\left(0\right)}={\left\|{\bar{A}}^{H}X\right\|}_{2}$. (ii) assign a, b, c and d very small values (ensure uninformative distributions). (iii) preset error tolerance ε and maximal iterative number $k_{\max}$.**Repetition** **while** (${\left\|p^{\left(k+1\right)}-p^{\left(k\right)}\right\|}_{2}/{\left\|p^{\left(k\right)}\right\|}_{2}>\varepsilon$ or $k<k_{\max}$) **do** (i) compute $\mu_{p}$ and $\Sigma_{p}$ according to (26) and (27). (ii) compute 〈Σ〉 and 〈α〉 according to (31) and (36).  (iii) Regard γ as $p^{\left(k\right)}$. (iv) k=k+1. **end while**
**Refinement** (i) use the final p to obtain $\theta^{est1}$. (ii) use the final $\mu_{p}$ to solve λ according to (41).  (iii) obtain $\theta^{est2}$ according to (42).**Output** The final DOA values.

Compared with the canonical SBL method, BSBL achieves better sparsity performance and lower computational complexity. According to the maximal number of complex multiplications, the complexity of BSBL, dominated by (26) (or rather $\Phi \langle \Sigma \rangle \Phi^{H}$), is $O(M^{2}NL^{3})$, less than $O(M^{3}L^{3})$ of the canonical SBL method.

However, BSBL will still suffer heavy computational burdens when L or M is large. Therefore, we must seek some techniques to reduce computational complexity.

### 4.2. BSBL-APPR

Obviously, the large complexity is mainly caused by high-dimensional matrix operations that contain massive useless zero (or near-zero) operations. Automatically, the simplest perspective is to exploit some approximation to shrink the matrix dimensions, generating the first faster version called BSBL-APPR. 

Recalling the entire algorithm, the high dimensions are essentially yielded by computing (26), (27), and (37). For $\mu_{p}$ in (26), the corresponding approximation is
(43)$$\begin{array}{ll} \mu_{p} & =\langle \Sigma \rangle \Phi^{H}{\left({\langle \alpha \rangle }^{-1}I_{NL}+\Phi \langle \Sigma \rangle \Phi^{H}\right)}^{-1}x\\ & =(\langle \Gamma^{-1}\rangle ⊗B){\left(\bar{A}⊗I_{L}\right)}^{H}[{\langle \alpha \rangle }^{-1}I_{NL}+(\bar{A}⊗I_{L})(\langle \Gamma^{-1}\rangle ⊗B){\left(\bar{A}⊗I_{L}\right)}^{H}]-1vec(X^{T})\\ & \approx (\langle \Gamma^{-1}\rangle {\bar{A}}^{H}⊗B)[{\left({\langle \alpha \rangle }^{-1}I_{N}+\bar{A}\langle \Gamma^{-1}\rangle {\bar{A}}^{H}\right)}^{-1}⊗B]vec(X^{T})\\ & =[\langle \Gamma^{-1}\rangle {\bar{A}}^{H}{\left({\langle \alpha \rangle }^{-1}I_{N}+\bar{A}\langle \Gamma^{-1}\rangle {\bar{A}}^{H}\right)}^{-1}⊗B^{2}]vec(X^{T})\\ & =vec(B^{2}X^{T}{\left[\langle \Gamma^{-1}\rangle {\bar{A}}^{H}{\left({\langle \alpha \rangle }^{-1}I_{N}+\bar{A}\langle \Gamma^{-1}\rangle {\bar{A}}^{H}\right)}^{-1}\right]}^{T}). \end{array}$$
where $\Gamma^{-1}=diag(\gamma^{-1})$. The derivation process follows Kronecker–Product properties, i.e., ${\left(A⊗B\right)}^{H}=A^{H}⊗B^{H}$, (A⊗B)(C⊗D)=AC⊗BD, and $vec(ABC)=(C^{T}⊗A)$ vec(B). The approximation exactly holds if ${\langle \alpha \rangle }^{-1}=0$ or $B=I_{L}$. To be specific, the approximation is reasonable if high SNRs or low ow correlation coefficient levels are adopted. In fact, the two conditions (or at least one) are easy to meet in practice.

Likewise, for $\Sigma_{p}$ in (27), the approximation is
(44)$$\begin{array}{ll} \Sigma_{p} & =\langle \Sigma \rangle -\langle \Sigma \rangle \Phi^{H}{\left({\langle \alpha \rangle }^{-1}I_{NL}+\Phi \langle \Sigma \rangle \Phi^{H}\right)}^{-1}\Phi \langle \Sigma \rangle \\ & =\langle \Gamma^{-1}\rangle ⊗B-(\langle \Gamma^{-1}\rangle ⊗B){\left(\bar{A}⊗I_{L}\right)}^{H}{\left[{\langle \alpha \rangle }^{-1}I_{NL}+(\bar{A}⊗I_{L})(\langle \Gamma^{-1}\rangle ⊗B){\left(\bar{A}⊗I_{L}\right)}^{H}\right]}^{-1}(\bar{A}⊗I_{L})(\langle \Gamma^{-1}\rangle ⊗B)\\ & \approx \langle \Gamma^{-1}\rangle ⊗B-(\langle \Gamma^{-1}\rangle {\bar{A}}^{H}⊗B)[{\left({\langle \alpha \rangle }^{-1}I_{N}+\bar{A}\langle \Gamma^{-1}\rangle {\bar{A}}^{H}\right)}^{-1}⊗B](\bar{A}\langle \Gamma^{-1}\rangle ⊗B)\\ & =\langle \Gamma^{-1}\rangle ⊗B-\langle \Gamma^{-1}\rangle {\bar{A}}^{-1}{\left({\langle \alpha \rangle }^{-1}I_{N}+\bar{A}\langle \Gamma^{-1}\rangle {\bar{A}}^{H}\right)}^{-1}\bar{A}\langle \Gamma^{-1}\rangle ⊗B^{3}. \end{array}$$

Similarly, for $\bar{d}$ in (37), its approximation is
(45)$$\begin{array}{ll} \bar{d} & =d+\frac{1}{2}({\left\|x-\Phi \mu_{p}\right\|}_{2}^{2}+tr(\Sigma_{p}\Phi^{H}\Phi ))\\ & \approx d+\frac{1}{2}{\left\|x-\Phi \mu_{p}\right\|}_{2}^{2}+\frac{1}{2}tr\{(\langle \Gamma^{-1}\rangle ⊗B)({\bar{A}}^{H}\bar{A}⊗I)-[\langle \Gamma^{-1}\rangle {\bar{A}}^{-1}{\left({\langle \alpha \rangle }^{-1}I_{N}+\bar{A}\langle \Gamma^{-1}\rangle {\bar{A}}^{H}\right)}^{-1}\bar{A}\langle \Gamma^{-1}\rangle ⊗B^{3}]({\bar{A}}^{H}\bar{A}⊗I)\}\\ & =d+\frac{1}{2}{\left\|x-\Phi \mu_{p}\right\|}_{2}^{2}+\frac{1}{2}tr\{(\langle \Gamma^{-1}\rangle {\bar{A}}^{H}\bar{A}⊗B)-[\langle \Gamma^{-1}\rangle {\bar{A}}^{-1}{\left({\langle \alpha \rangle }^{-1}I_{N}+\bar{A}\langle \Gamma^{-1}\rangle {\bar{A}}^{H}\right)}^{-1}\bar{A}\langle \Gamma^{-1}\rangle {\bar{A}}^{H}\bar{A}⊗B^{3}]\}. \end{array}$$

After the approximation, BSBL-APPR is completed. Its concrete iterative steps are omitted here since the process is the same as BSBL, except for the calculation of $\mu_{p}$, $\Sigma_{p}$, and $\bar{d}$ by (43)–(45).

Since the approximation operations have been performed, the computational complexity of BSBL-APPR, dominated by $\bar{A}\langle \Gamma^{-1}\rangle {\bar{A}}^{H}$, is $O(M^{2}N)$ less than $O(M^{2}NL^{3})$, which theoretically verifies the higher efficiency of BSBL-APPR. 

Even so, the complexity of BSBL-APPR still seems to be intolerable when dense sampling is adopted, i.e., M is large enough. Additionally, there exist several operations with many zero (or near-zero) elements, e.g., $\bar{A}\langle \Gamma^{-1}\rangle {\bar{A}}^{H}$ and $\langle \Gamma^{-1}\rangle ⊗B$. For lower complexity, the most effective method is to decompose matrix operations into vector and even scalar operations so as to selectively avoid useless computation. Fortunately, a technique, named generalized approximate message passing (GAMB), exactly achieves that.

### 4.3. BSBL-GAMP

GAMP is a technique developed to solve approximate marginal posteriors with low complexity based on the central limit theorem [58]. To briefly explain the principle of the GAMP technique, (3) is rewritten in scalar form.
(46)$$x_{i}=\Phi_{i\cdot }p+n_{i},\forall i=1,\ldots ,NL.$$
where $x_{i}$ and $n_{i}$ are the i−th entries of x and n. Let $z_{i}=\Phi_{i\cdot }p$, $p={\left[p_{1},\ldots ,p_{j},\ldots ,p_{ML}\right]}^{T}$, and j=1,…,ML. Given the known measurement matrix Φ and the observed vector x, the objective is to obtain the estimation of p. 

Each $x_{i}$ is connected to $p_{j}$ by Φ, and vice versa. $x_{i}$ and $p_{j}$ are defined as the input node and the output node, respectively. The association between them is called an edge. Input nodes and output nodes pass messages to each other along the edges. The original message passing (MP) technique is to keep passing messages (i.e., probability distributions) with respect to $p_{j}$ until convergence. Based on MP, GAMP is the extension for low complexity since it passes only important messages that mainly affect the approximated marginal posteriors of p. 

Overall, the GAMP technique is selected to speed up our algorithm for two motivations. (i) It passes messages from one node to another, enforcing scalar operations with low complexity. (ii) It can also compute approximate marginal posteriors of p, which allows GAMP to be embedded into the proposed BSBL.

To apply GAMP to our used block-sparse data model, we must derive it again. Along the line of the common procedures of GAMP, there are two important approximate marginal posteriors to consider. One is
(47)$$p(z_{i}|x,\upsilon_{i}^{z},\tau_{i}^{z},\eta )=\frac{p(x_{i}|z_{i},\eta )CN\left(z_{i}|\upsilon_{i}^{z},\tau_{i}^{z}\right)}{\int_{z}p(x_{i}|z_{i},\eta )CN\left(z_{i}|\upsilon_{i}^{z},\tau_{i}^{z}\right)dz}.$$
for approximating $p(z_{i}|x,\eta )$, i=1,2,…,NL, where η={γ,α}. $\upsilon_{i}^{z}$ and $\tau_{i}^{z}$ are quantities to be updated. The other is
(48)$$p(p_{j}|x,\upsilon_{j}^{p},\tau_{j}^{p},\eta )=\frac{p(p_{j}|\eta )CN(p_{j}|,\upsilon_{j}^{p},\tau_{j}^{p})}{\int_{p}p(p_{j}|\eta )CN(p_{j}|,\upsilon_{j}^{p},\tau_{j}^{p})dp}.$$
for approximating $p(p_{j}|x,\eta )$, j=1,2,…,ML, where $\upsilon_{j}^{p}$ and $\tau_{j}^{p}$ are quantities to be updated. Given the model and priors, it is easy to obtain $p(x_{i}|z_{i},\eta )=CN(x_{i}|z_{i},\alpha^{-1})$ and $p(p_{j}|\eta )=CN(p_{j}|0,\gamma_{j_{1}}^{-1}\beta_{j_{2},j_{2}})$, where $\beta_{j_{2},j_{2}}$ is the $j_{2}-th$ row, $j_{2}-th$ is the column element of B, and $(j_{1},j_{2})=\{\begin{array}{c} (m,L),ifj=mL\\ (\left\lceil j/L\right\rceil ,\mathrm{mod}(j,L)),else \end{array}$, ∀m=1,…M. $j_{1}=1,\ldots ,M$, $j_{2}=1,\ldots ,L$. Similar to (7), (47) and (48) are easily identified as Gaussian distributions.
(49)$$p(z_{i}|x,\upsilon_{i}^{z},\tau_{i}^{z},\eta )=CN(z_{i}|\mu_{i}^{z},\phi_{i}^{z}).$$
(50)$$p(p_{j}|x,\upsilon_{j}^{p},\tau_{j}^{p},\eta )=CN(p_{j}|\mu_{j}^{p},\phi_{j}^{p}).$$
where $\mu_{i}^{z}=\frac{\upsilon_{i}^{z}+\alpha x_{i}\tau_{i}^{z}}{1+\alpha \tau_{i}^{z}}$ and $\phi_{i}^{z}=\frac{\tau_{i}^{z}}{1+\alpha \tau_{i}^{z}}$ are the mean and variance of $p(z_{i}|x,\upsilon_{i}^{z},\tau_{i}^{z},\eta )$, while $\mu_{j}^{p}=\frac{\upsilon_{i}^{p}\beta_{j_{2},j_{2}}}{\beta_{j_{2},j_{2}}+\gamma_{j_{1}}\tau_{j}^{p}}$ and $\phi_{j}^{p}=\frac{\tau_{j}^{p}\beta_{j_{2},j_{2}}}{\beta_{j_{2},j_{2}}+\gamma_{j_{1}}\tau_{j}^{p}}$ are the mean and variance of $p(p_{j}|x,\upsilon_{j}^{p},\tau_{j}^{p},\eta )$. Please refer to Appendix E for details of the derivation. Then, it is required to determine two scalar functions denoted as $g_{in}(\cdot )$ and $g_{out}(\cdot )$, where $g_{in}(\cdot )$ is equal to the posterior mean $\mu_{j}^{p}$ of $p_{j}$, i.e.,
(51)$$g_{in}(\upsilon_{j}^{p},\tau_{j}^{p},\eta )=\mu_{j}^{p}=\frac{\upsilon_{j}^{p}\beta_{j_{2},j_{2}}}{\beta_{j_{2},j_{2}}+\gamma_{j_{1}}\tau_{j}^{p}}.$$

The corresponding posterior variance is
(52)$$\tau_{j}^{p}\frac{\partial }{\partial \upsilon_{j}^{p}}g_{in}(\upsilon_{j}^{p},\tau_{j}^{p},\eta )=\phi_{j}^{p}=\frac{\tau_{j}^{p}\beta_{j_{2},j_{2}}}{\beta_{j_{2},j_{2}}+\gamma_{j_{1}}\tau_{j}^{p}}.$$

$g_{out}(\cdot )$ satisfies
(53)$$g_{out}(\upsilon_{i}^{z},\tau_{i}^{z},\eta )=\frac{1}{\tau_{i}^{z}}(\mu_{i}^{z}-\upsilon_{i}^{z})=\frac{1}{\tau_{i}^{z}}(\frac{\upsilon_{i}^{z}+\alpha x_{i}\tau_{i}^{z}}{1+\alpha \tau_{i}^{z}}-\upsilon_{j}^{z}).$$

The corresponding posterior variance is
(54)$$\tau_{i}^{z}\frac{\partial }{\partial \upsilon_{i}^{z}}g_{out}(\upsilon_{i}^{z},\tau_{i}^{z},\eta )=\frac{-\alpha \tau_{i}^{z}}{1+\alpha \tau_{i}^{z}}.$$

So far, the derivation of GAMP is completed. Note that the variance of (52) becomes a vector (equivalent to a diagonal matrix), while $\Sigma_{p}$ of BSBL (or BSBL-APPR) is still a normal matrix. From this perspective, BSBL-GAMP is generated by the most thorough approximation, resulting in the least complexity. As shown in Algorithm 2, the GAMP algorithm is summarized.

Intuitively, only $\mu_{p}$ and diagonalized $\Sigma_{p}$ could be updated in the GAMP algorithm. For hyperparameter α, its update process will still lead to relatively large complexity if computed by (36). Here, α is rewritten as
(55)$$\langle \alpha \rangle =\frac{c+NL/2}{d+\frac{1}{2}{\left\|x-\Phi \mu_{p}\right\|}_{2}^{2}+\sum_{i=1}^{M}{\left(\Sigma_{p}\right)}_{i\cdot }{\left(\Phi^{H}\Phi \right)}_{\cdot i}}.$$

Overall, the second faster version, called BSBL-GAMP, is eventually yielded by embedding the GAMP algorithm into BSBL. Specific steps are summarized in Algorithm 3.
**Algorithm 2.** The proposed GAMP algorithm.**Initialization** $(i)\;\mathrm{set}\;{\hat{s}}_{i}=0$,k=0, input γ and α. $(\mathrm{ii})\;\mathrm{use}\;(22)\;\mathrm{to}\;\mathrm{compute}\;\mu_{p}={\left\{\mu_{j}^{p}\right\}}_{j=1}^{NL}$. $(\mathrm{iii})\;\mathrm{use}\;(23)\;\mathrm{to}\;\mathrm{compute}\;diag(\Sigma_{p})={\left\{\phi_{j}^{p}\right\}}_{j=1}^{NL}$. (iv) preset error tolerance ε.**Repetition**∀i=1,…,NL, j=1,…,ML$,\;j_{1}=1,\ldots ,M$$,j_{2}=1,\ldots ,L$. $\left(i)\;{\hat{z}}_{i}=\sum_{j}\Phi_{i,j}\mu_{j}^{p}(k\right)$. $\left(\mathrm{ii})\;\tau_{i}^{z}=\sum_{j}\Phi_{i,j}^{2}\phi_{j}^{p}(k\right)$. $(\mathrm{iii})\;\upsilon_{i}^{z}={\hat{z}}_{i}-\tau_{i}^{z}{\hat{s}}_{i}$. $(\mathrm{iv})\;{\hat{s}}_{i}=[(\upsilon_{i}^{z}+\alpha x_{i}\tau_{i}^{z})/(1+\alpha \tau_{i}^{z})-\upsilon_{i}^{z}]/\tau_{i}^{z}$. $\left(v)\;\tau_{i}^{s}=\alpha /(1+\alpha \tau_{i}^{z}\right)$. $(\mathrm{vi})\;\tau_{j}^{p}={\left(\sum_{i}\Phi_{i,j}^{2}\tau_{i}^{s}\right)}^{-1}$. $\left(\mathrm{vii})\;\upsilon_{j}^{p}=\mu_{j}^{p}(k)+\tau_{j}^{p}(\sum_{i}\Phi_{i,j}{\hat{s}}_{i}\right)$. $\left(\mathrm{viii})\;\mu_{j}^{p}(k+1)=\upsilon_{j}^{p}\beta_{j_{2},j_{2}}/(\beta_{j_{2},j_{2}}+\gamma_{j_{1}}\tau_{j}^{p}\right)$. $\left(\mathrm{ix})\;\phi_{j}^{p}(k+1)=\tau_{j}^{p}\beta_{j_{2},j_{2}}/(\beta_{j_{2},j_{2}}+\gamma_{j_{1}}\tau_{j}^{p}\right)$. (x) k=k+1.**Terminate** ${\left\|\mu_{j}^{p}(k+1)-\mu_{j}^{p}(k)\right\|}_{2}\leq \varepsilon$.**Output**$\;\mathrm{The}\;\mathrm{final}\;\mu_{p}$ $\mathrm{and}\;\Sigma_{p}$.

**Algorithm 3.** The proposed GAMP-BSBL algorithm.
**Initialization.**
 (i) set the first iterative number k=0, error tolerance ε, maximal iterative number $k_{\max}$. (ii) set $p^{\left(0\right)}={\left\|{\bar{A}}^{H}X\right\|}_{2}$. (iii) assign a, b, c and d very small values. (iv) compute γ and α with (31) and (55), respectively.
**Repetition.**
 (i) compute $\mu_{p}$ and $\Sigma_{p}$ according to the above GAMP. (ii) compute γ and α according to (31) and (55), respectively. (iii) update k=k+1. (iv) regard γ as $p^{\left(k\right)}$.**Terminate** ${\left\|p^{\left(k+1\right)}-p^{\left(k\right)}\right\|}_{2}/{\left\|p^{\left(k\right)}\right\|}_{2}\leq \varepsilon$ or $k=k_{\max}$

**Refinement**
 (i) use the final p to obtain $\theta^{est1}$. (ii) use the final $\mu_{p}$ to solve λ according to (41). (iii) obtain $\theta^{est2}$ according to (42).**Output** the final DOA values.

Theoretically, BSBL-GAMP contains only simple multiplication of vectors and linear operations, so it is undoubtedly the fastest algorithm. To be specific, its complexity dominated by (55) is O(ML), much less than $O(M^{2}NL^{3})$ of BSBL or $O(M^{2}N)$ of BSBL-APPR. For comparison, the computational complexity of all the narrowband algorithms involved in this paper is summarized in Table 2. Generally, M≫N,L holds; thus, BSBL-GAMP obviously has the least computational complexity. In contrast, BSBL-APPR seems to be also satisfactory since its complexity is smaller than others, except for IC-SPICE and RVM-DOA. Moreover, the complexity of BSBL is moderate.

### 4.4. BSBL for Wideband Sources

Although BSBL, BSBL-APPR, and BSBL-GAMP are developed, they are only applicable in the case of narrowband sources. For wideband cases, we must extend BSBL further.

A way to deal with wideband sources is to separate the wideband spectrum into independent narrowband ones. Without loss of generality, (3) can be rewritten as the special case at the j−th frequency point $f_{j}$, ∀j=1,…J, i.e.,
(56)$$x_{j}=\Phi_{j}p_{j}+n_{j}.$$

In this model, it is worth emphasizing that we only care about the locations of non-zero elements in pj rather than the concrete values because different pj theoretically indicate the same location of some source. Consequently, considering all the frequency points, different pj can be unified as q so that
(57)y=Ψq+n¯.
where y=[x1T,…,xJT]T∈ℂNLJ×1, Ψ=[Φ1T,…,ΦJT]T ∈ℂNLJ×ML, and n¯=[n1T,…,nJT]T∈ℂNLJ×1. Similar to the aforementioned derivation of BSBL, (58)−(65) are yielded as follows.
(58)μq=〈Σw〉ΨH(〈αw〉−1INLJ+Ψ〈Σw〉ΨH)−1y.
(59)Σq=〈Σw〉−〈Σw〉ΨH(〈αw〉−1INLJ+Ψ〈Σw〉ΨH)−1Ψ〈Σw〉.
(60)a¯w=aw+12.
(61)b¯w,m=bw+12‖(B−1+1L×L−IL)⊙(μqmμqmH)‖1,1+12diag(B−1⊙Σqm).
(62)〈γmw〉=a¯wb¯w,m.
(63)c¯w=cw+NLJ2.
(64)d¯w=dw+12(‖y−Ψμq‖22+tr(ΣqΨHΨ)).
(65)〈αw〉=c¯wd¯w.

To be distinguished from BSBL in narrowband cases, the diffetent parameters in wideband cases are with superscript or subscript w and q. In particalur, $\Sigma_{w}=diag({\left(\gamma^{w}\right)}^{-1})⊗B$ is the variance of variable q, where $\gamma^{w}={\left[\gamma_{1}^{w},\ldots ,\gamma_{M}^{w}\right]}^{T}$. In wideband cases, the off-grid correction is the same as BSBL, except for (41), which is replaced by
(66)$$\lambda_{w}⊙\mu_{q}={\left[{\left(\Psi^{\prime }\right)}^{H}\Psi^{\prime }\right]}^{-1}{\left(\Psi^{\prime }\right)}^{H}(y-\Psi \mu_{q}).$$

So far, BSBL for wideband sources has been completed. The specific process is summarized in Algorithm 4.
**Algorithm 4.** BSBL for wideband sources.**Initialization**.  (i) set k=0, $q^{\left(0\right)}={\left\|\sum_{j=1}^{J}{\bar{A}}_{j}^{H}X_{j}\right\|}_{2}/J$. (ii) set $a_{w}$, $b_{w}$, $c_{w}$ and $d_{w}$ very small values. (iii) preset error tolerance ε and maximal iterative number $k_{\max}$.**Repetition** **while** (${\left\|q^{\left(k+1\right)}-q^{\left(k\right)}\right\|}_{2}/{\left\|q^{\left(k\right)}\right\|}_{2}>\varepsilon$ or $k<k_{\max}$) **do**:  (i) compute $\mu_{q}$ and $\Sigma_{q}$ according to (58) and (59). (ii) compute $\langle \Sigma_{w}\rangle$ and $\langle \alpha_{w}\rangle$ according to (62) and (65). (iii) regard $\gamma^{w}$ as $q^{\left(k\right)}$. (iv) k=k+1. **end while**
**Refinement** (i) Use the final q to obtain $\theta^{est1}$. (ii) Use the final $\mu_{q}$ to solve $\lambda_{w}$ according to (66). (iii) Obtain $\theta^{est2}$ according to (42).**Output** The final DOA values.

## 5. Numerical Simulation

In this section, the superiority of our proposed algorithms will be proven comprehensively through three subsections, including extensive simulations. For simplicity, the proposed BSBL, BSBL-APPR, and BSBL-GAMP are collectively referred to as BSBLs. In the first and second subsections, the narrowband and wideband estimation performance of various algorithms is evaluated comprehensively. In the third subsection, the in-depth analysis of Bayesian performance is completed by comparison with other off-grid SBL-based methods.

### 5.1. Estimation Performance for Narrowband Sources

In this subsection, estimation performance is evaluated by the Root-Mean-Square Error (*RMSE*) expressed as
(67)$$RMSE=\sqrt{\frac{1}{M_{c}K}\sum_{m_{c}=1}^{M_{c}}\sum_{k=1}^{K}{\left({\hat{\theta }}_{m_{c},k}-\theta_{k}\right)}^{2}}.$$
where $M_{c}$ is the Monte Carlo number and K is the number of sources. ${\hat{\theta }}_{m_{c},k}$ is the estimation value for the k−th source in the $m_{c}-th$ trial and $\theta_{k}$ is the true angle of the k−th source. For clarity, we introduce four canonical SMV algorithms, i.e., $l_{1}$−SVD [20], $l_{1}$−SRACV [21], IC-SPICE [22], SBL [31], SS-ANM [25], and StrucCovMLE [26], as comparisons in following simulations. Before that, unless otherwise stated, baseline simulation conditions are SNR=20dB, K=3 temporally correlated sources with a random DOA set $\left\{-20.53^{\circ },10.10^{\circ },43.01^{\circ }\right\}$, the number of sensors is N=8, the number of snapshots is L=4, the grid interval is $1^{\circ }$, the number of grids is M=180, the Monte Carlo number is $M_{c}=200$, and the temporal correlation coefficient is β=0. 

**Remark** **3.**β=0 *and* SNR=20 dB *(or at least one) are provided to ensure that BSBL-APPR performs normally.*

Simulation 1 tests the ability of various algorithms to suppress temporal correlation. Specifically, the amplitudes and phases of temporally correlated coefficients uniformly vary from [0,1] and [0,2π], respectively. The results are shown in Figure 2. Obviously, the *RMSE*s of BSBLs persistently remain low and fluctuate only slightly with the correlation coefficient varying, while others just struggle when the correlation coefficient is large. Particularly, IC-SPICE is able to impair the influence of temporal correlation to some extent, but it is still at a loss if the correlated level is high. Overall, the simulation results fully live up to our expectations that BSBLs are able to suppress temporal correlation effectively, which confirms that the considered temporal correlation modeled in the block-sparse model indeed plays an important role in improving the robustness of temporal correlation. 

Simulation 2 examines the dependence on the number of snapshots. In Figure 3, all the sparsity-based algorithms are collectively robust to various snapshots. In other words, all the algorithms seem to achieve SSR with only a few snapshots. Despite this, BSBLs are still commendable due to the realized lowest *RMSE*s. In fact, SBL itself enables finding global minima and smoothing out numerous local minima in some cases with a few snapshots [48]. For SBL-based methods, BSBLs undoubtedly inherit their advantages. Additionally, the used hierarchical priors can improve sparsity performance so that BSBLs perform best. Thus, BSBLs enable high estimated precision with a few snapshots.

Simulation 3 focuses on the *RMSE* performance with respect to SNRs. Apparently, in Figure 4, all the algorithms work well if high SNRs are adopted, while only BSBLs maintain fewer *RMSE*s at low SNRs. The results can be explained by the fact that the sparsity-based algorithms rely on high SNRs to some extent, but SBL can reduce the dependency. Taking IC-SPICE as an example, it can achieve efficient iterative optimization under the condition of high SNRs but cannot seek the right global minima or even trap some fixed local optima. The original reason for this is that the used covariance matrix and the updated parameters have large errors with ideal ones. However, SBL seems to work normally owing to convergence guarantee and gradual optima under the condition of existing data errors. Surprisingly, the proposed BSBLs have a similar ability in some way. On the whole, BSBLs are preferable, especially under the condition of low SNRs.

Simulation 4 investigates the *RMSE* performance with respect to the number of sensors. Here, this simulation is executed at the number of snapshots L=20 rather than L=4 since $l_{1}$−SVD cannot work normally when the number of sensors exceeds the number of snapshots. Intuitively, in Figure 5, the *RMSE* performance of BSBLs is still excellent, although BSBLs are inferior to IC-SPICE at N=4. In fact, the results are related to the ability to solve underdetermined DOA estimation problems. SBL still seems to find the sparsest solutions, although the restricted isometry property (RIP) is not satisfied [48]. When the number of sensors is not large enough, i.e., the solution to be solved is not sparse enough, SBL still tries its best to realize global minima. Thus, BSBLs can handle underdetermined cases efficiently. In other words, BSBLs are able to realize SSR effectively on the condition of a few sensors.

Simulation 5 tests the adaptability to wide grid intervals. In Figure 6, the results illustrate that both BSBLs and IC−SPICE gain highly accurate estimation values at refined grids, but only BSBLs reluctantly adapt to wide grid intervals, although all the algorithms suffer hardship at coarse grids. The coarse girds compel relatively large errors of the pre-estimated values, so the refined values have a larger bias. In the proposed BSBLs, the grid refinement has an effect on reducing the bias at each iteration. Therefore, BSBLs adapt to coarse grids to some extent.

Simulation 6 examines the *RMSE* performance with respect to the number of sources. The different conditions are as follows: the DOA sets of sources are selected from $\left[-90^{\circ },90^{\circ }\right]$ randomly, and L=10 is chosen to ensure the normal operation of $l_{1}$−SVD. In Figure 7, all the *RMSE*s drop rapidly, but BSBLs seem to slow down the pace of performance degradation, which implies that BSBLs have the potential to locate more sources. In fact, the results are another proof of the excellent underdetermined DOA estimation ability of BSBLs in Simulation 4. More sources and fewer sensors play similar roles in decreasing the sparse level in sparse recovery theory, and BSBLs handle this case efficiently. 

### 5.2. Estimation Performance for Wideband Sources

For comparison, JLZA−DOA [23], W−SpSF [24], W−SBL [36], GS−WSpSF [27], and ANM [28] are introduced. On behalf of BSBLs, only BSBL is adopted in the last two subsections for simplicity.

Simulation 7 tests the spectral performance of the above algorithms. Its conditions are two uncorrelated chirps with angles of $-10^{\circ }$ and $25^{\circ }$ with a center frequency of 1 kHz and a bandwidth of 400 Hz from 0.8 kHz to 1.2 kHz, SNR=20dB, the number of sensors is N=8, the number of snapshots is L=4, and the grid interval is $1^{\circ }$. Intuitively, in Figure 8, JLZA−DOA and W−SpSF arise explicit sidelobes around their spikes, while GS−WSpSF, ANM, W−SBL, and BSBL are excellent since their spectra are almost without sidelobes and the spikes are sharp. It is worth noting that the spikes of W−SBL seem to be very low at some frequency points. Specifically, W-SBL fluctuates intensely with varied frequencies, i.e., the signal energy non-uniformly leaks between different frequencies. GS−WSpSF and ANM are better; at least their spikes are visible and apparent over the range of all the frequency points. The proposed BSBL has the highest spikes and varies little over the range of frequencies. Thus, three conclusions are drawn. (1) When sparse recovery is adopted, BSBL can ensure that equal energy is assigned between different frequency bins. (2) BSBL realizes the highest spikes and shows the best convergence effect, i.e., ensuring global minima.

Simulation 8 examines the *RMSE* performance with respect to the number of sensors. The condition set is the same as baseline conditions, except for wideband sources. As expected, BSBL achieves excellent *RMSE* performance in Figure 9. Over the whole range of the number of snapshots, BSBL shows overwhelming advantages and outperforms others patently. In fact, DOA estimation for wideband sources is difficult, and one of the main reasons is many algorithms fail to realize accuracy estimation over the whole range of frequency bins. Simulation 7 shows the special ability of BSBL to overcome this problem, and Simulation 8 seems to verify it again. On the one hand, BSBL for wideband sources maintains the superiority of narrowband BSBL, so it can obtain excellent estimation performance. On the other hand, BSBL extends the advantages of SBL to wideband sources, i.e., achieving robust sparse recovery with only a few snapshots for all the sources over the whole frequency band.

Simulation 9 tests the *RMSE* performance with respect to SNRs. In Figure 10, BSBL outperforms others and improves well when SNRs increase. It is worth noting that the results are different from the narrowband ones in Simulation 2. To be specific, BSBL seems to fluctuate intensely with the varied SNRs. BSBL for wideband sources cannot work well enough compared to narrowband source cases. We carefully analyze the reasons and find that BSBL cannot realize ensuring global minima at each frequency bin for sparse recovery. Despite this, the defects cannot obscure the virtues. BSBL still achieves impressive performance for wideband sources. 

### 5.3. Analysis of Sparse Bayesian Performance

According to the common perception of SBL, the elaborate Bayesian framework with substantial priors is regarded as a characteristic to enhance sparsity well because priors play a role of regularization in sparse recovery [55,59]. Here, we abstract several Bayesian frameworks from several off-grid SBL-based algorithms, such as RVM−DOA [37], RV−ON−SBL [47], ON−SBLRVM [43], SBLMC [39], and HSL [46], for comparison and analysis. As shown in Figure 11, RVM−DOA, RV−ON−SBL, and ON−SBLRVM are only imposed on Gaussian priors that have been proven of poor Bayesian performance. The complicated Bayesian frameworks of HSL and BSBL are the same, except for the priors assigned to the unknown variables; thus, they may perform equally well. SBLMC has the most elaborate Bayesian framework composed of sufficient priors, so its sparsity performance will be perfect theoretically.

To confirm the above conjectures, two more simulations are performed to test the estimation performance of these algorithms.

Simulation 10 tests the *RMSE* performance with respect to the number of snapshots. The conditions are the same as the baseline conditions. In Figure 12, the *RMSE* performance of RVM−DOA, RV−ON−SBL, and ON−SBLRVM are expectedly worse than others, but SBLMC seems to not meet our expectations, while BSBL and HSL achieve preeminent *RMSE* performance. BSBL and HSL with moderately elaborate Bayesian frameworks outperform others, including the SBLMC with the most elaborate one. The result seems to violate the rule of Bayesian learning, which will be explained in the following text.

Simulation 11 tests the *RMSE* performance with respect to SNRs. In Figure 13, the proposed BSBL still seems to work best. It is worth noticing that the advantages of BSBL are not obvious, especially when SNRs are low. For all the SBL-based methods, BSBL has shown no more advantages than others on the condition of low SNRs because hierarchical priors improve sparsity if, and only if, SNRs are high. To be specific, Bayesian learning is able to find global minima, even at low SNRs, but the parameters yielded by hierarchical priors seem to update well only if SNRs are high.

Based on the simulation results, it can be seen that the complicated Bayesian frameworks, i.e., Ⅱ, Ⅲ, and Ⅳ in Figure 10, indeed achieve more excellent Bayesian performance than the canonical one, i.e., Ⅰ in Figure 11. However, SBLMC with the most elaborate Bayesian framework has not met our initial expectation, which can be explained by the fact that (i) SBL belongs to machine learning, so the Bayesian framework with too many priors will yield massive iterative hyperparameters, leading to overfitting during the iterative process. (ii) SBLMC is developed in the presence of mutual coupling; thus, the involved additional hyperparameter iteration is bound to affect the key parameters related to DOA estimation.

It is worth emphasizing that BSBL achieves slightly better estimation performance than HSL. The result indicates that the indirectly induced Student’s t priors, generated by Gaussian and Gamma priors, indeed express excellent sparsity performance. In fact, Student’s t priors have preferable sparsity-inducing performance, which has been mentioned in [55,56].

Overall, the above three subsection simulation results sufficiently demonstrate the superiority of BSBLs. Understandably, BSBL leverages hierarchical Gaussian and Gamma priors and uses VBI to complete the Bayesian inference so as to construct the corresponding iterative algorithm. Theoretically, the superiority is guaranteed by (1) indirect Student’s t-distributions, which have an excellent sparsity-inducing ability [56], and (2) variational approximation for Bayesian inference shows better performance than maximum posterior (MAP) estimation adopted in many SBL-based methods [60]. In addition, two approximation operations have achieved impressive running efficiency beyond many state-of-the-art methods. Moreover, BSBL still performs well in wideband cases and outperforms other algorithms in smoothing the spectrum peaks and super resolutions. Last but not least, BSBL has suppressed temporal correlation efficiently owing to its tactful algorithm design.

## 6. Conclusions

In this paper, we develop a DOA estimator (i.e., BSBL) based on sparse Bayesian learning with hierarchical priors. Due to the unacceptable computational complexity caused by the vectorization of the MMV model, two approximation operations are creatively introduced, thereby yielding two faster versions of BSBL, i.e., BSBL-APPR and BSBL-GAMP. As expected, all the proposed BSBLs (including BSBL, BSBL-APPR, BSBL-GAMP) achieve excellent estimation performance. For narrowband source estimation, BSBLs show perfect sparsity performance owing to the designed hierarchical priors. Further, BSBLs inherit and even extend the advantages of SBL, such as sparse signal recovery guarantee, less dependency on numerous snapshots or high SNRs, and the ability to handle underdetermined DOA estimation. Moreover, BSBLs enable robustness to temporally correlated sources and adaptability to coarse grids, which owes to the considered temporal correlation and the used grid refinement. For wideband source estimation, BSBL almost maintains huge advantages, i.e., realizing highly accurate estimation among the whole frequency band, while others suffer performance reduction to varying degrees. However, in wideband cases, BSBL cannot retain the good performance as in narrowband cases if low SNRs are adopted, which is our goal to solve in the next study. For Bayesian performance, BSBL with a moderately elaborate Bayesian framework realizes the best estimation performance. Furthermore, BSBL can balance both sparsity and complexity. Specifically, BSBL achieves sharp spectrum spikes and avoids overfitting produced by too many parameters.

Overall, the proposed BSBLs tactfully combine the hierarchical priors and the block-sparse model that contribute much to complexity reduction, which is never achieved by other SBL-based methods. Moreover, BSBLs retain and extend the advantages of SBL. Most importantly, BSBL is more practical and applicable when sources are temporally correlated or wideband. Despite this, BSBL seems not to be perfect because its performance suffers a loss at low SNRs in wideband cases to some extent. Anyway, the proposed BSBLs are worth recommendation and praise.

## Figures and Tables

**Figure 1 sensors-24-02336-f001:**
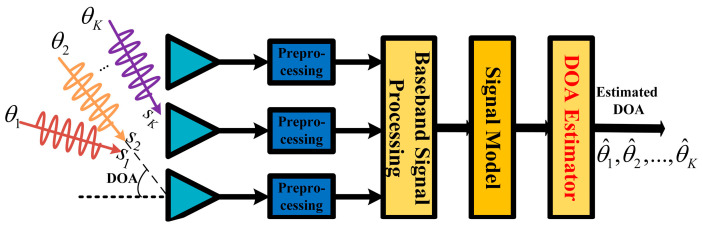
Flowchart of radar signal processing for DOA estimation.

**Figure 2 sensors-24-02336-f002:**
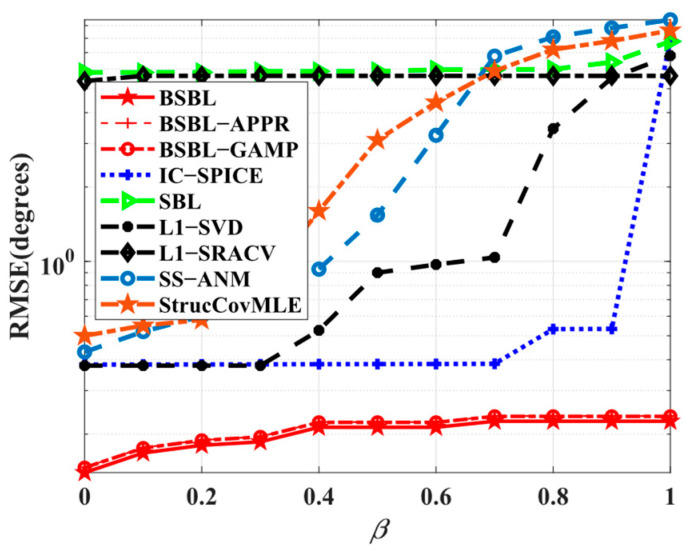
*RMSE* versus correlation coefficient.

**Figure 3 sensors-24-02336-f003:**
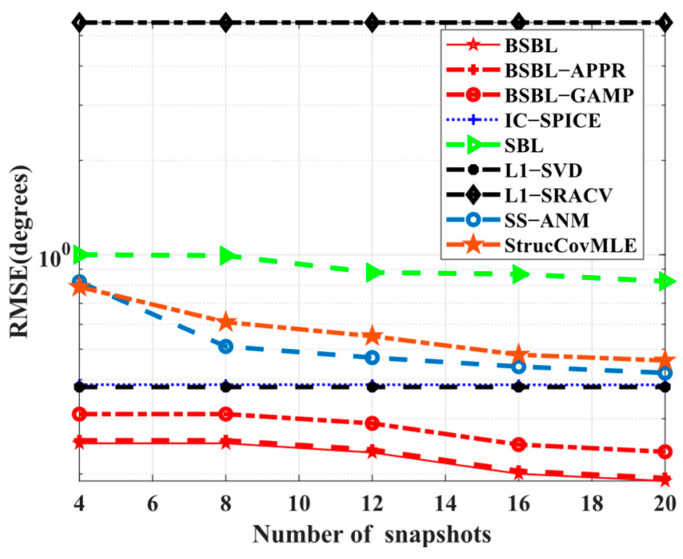
*RMSE* versus number of snapshots.

**Figure 4 sensors-24-02336-f004:**
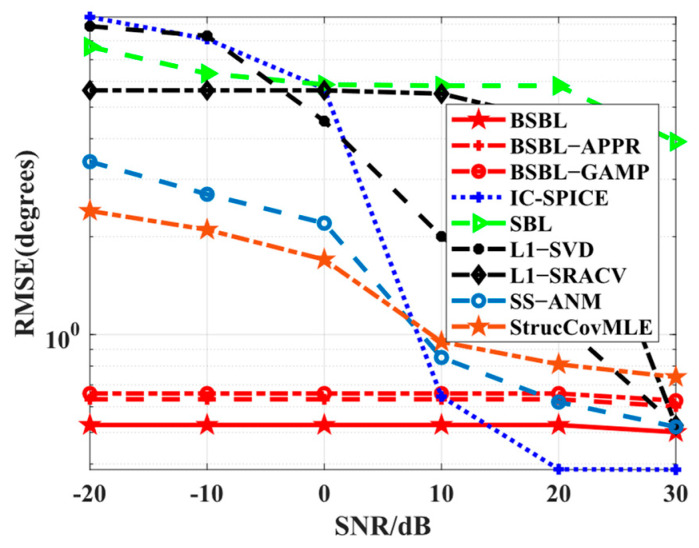
*RMSE* versus SNR.

**Figure 5 sensors-24-02336-f005:**
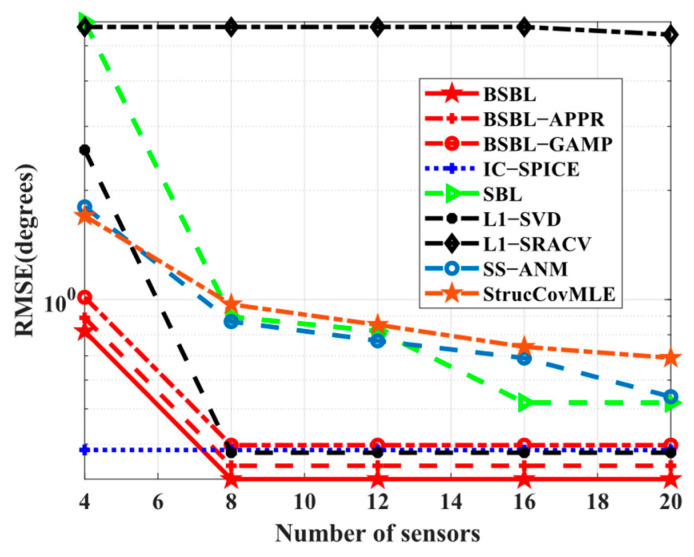
*RMSE* versus number of sensors at L=20.

**Figure 6 sensors-24-02336-f006:**
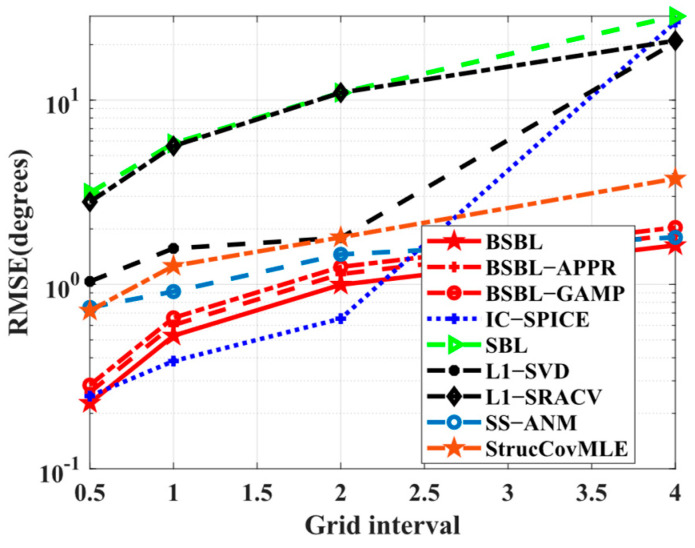
*RMSE* versus grid interval.

**Figure 7 sensors-24-02336-f007:**
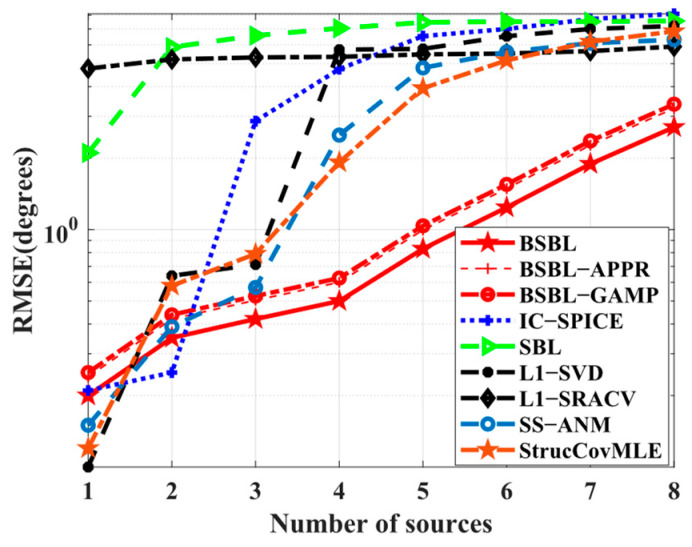
*RMSE* versus number of sources.

**Figure 8 sensors-24-02336-f008:**
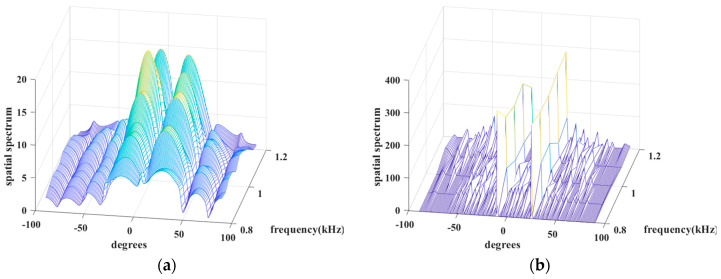
Spatial spectrum versus degree and frequency of (**a**) JLZA−DOA, (**b**) W−SpSF, (**c**) W−SBL, (**d**) GS−WSpSF, (**e**) ANM, (**f**) BSBL.

**Figure 9 sensors-24-02336-f009:**
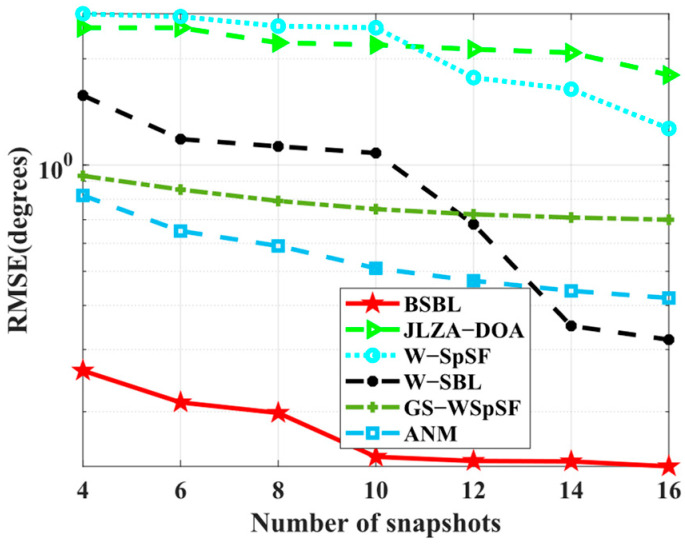
*RMSE* versus number of snapshots.

**Figure 10 sensors-24-02336-f010:**
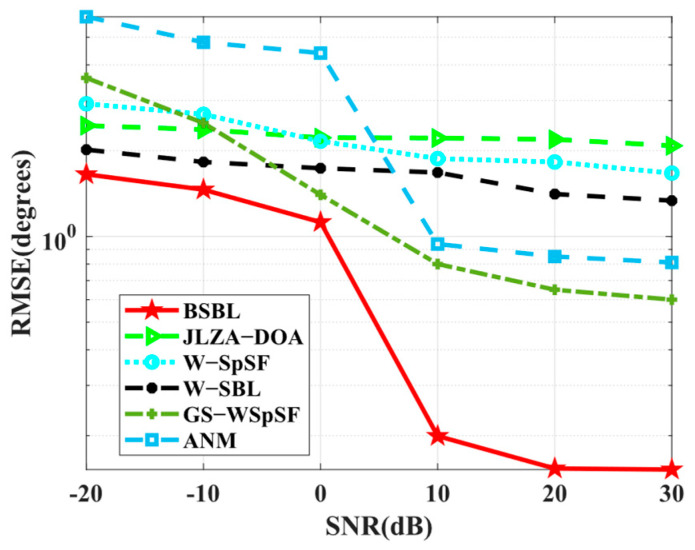
*RMSE* versus SNR.

**Figure 11 sensors-24-02336-f011:**
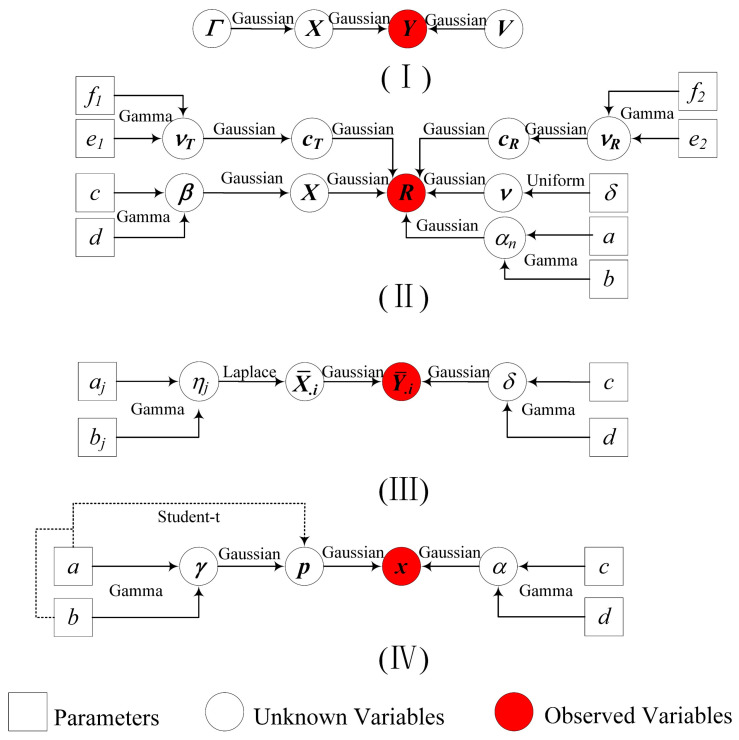
Directed acyclic graph representing the Bayesian frameworks of (**Ⅰ**) RVM−DOA, RV−ON−SBL, ON−SBLRVM, (**II**) SBLMC, (**III**) HSL, (**IV**) BSBL.

**Figure 12 sensors-24-02336-f012:**
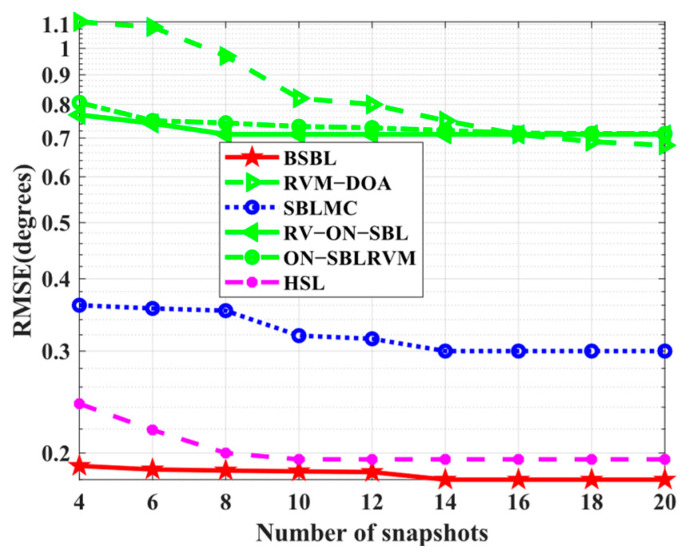
*RMSE* versus number of snapshots.

**Figure 13 sensors-24-02336-f013:**
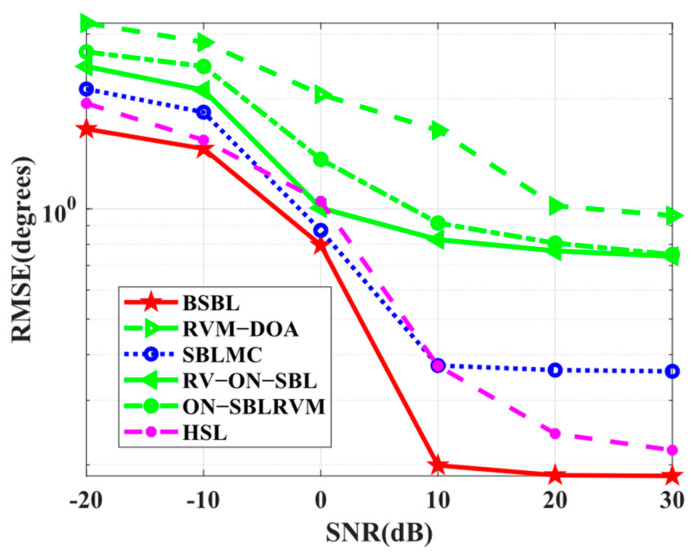
*RMSE* versus SNR.

**Table 1 sensors-24-02336-t001:** List of notations.

Symbol	Description
${\left(\cdot \right)}^{T}$	Transpose
${\left(\cdot \right)}^{*}$	Complex conjugate
${\left(\cdot \right)}^{H}$	Hermitian transpose
ℝ	Set of real numbers
ℂ	Set of complex numbers
CN(x|μ,Σ)	x Obeys a complex Gaussian distribution with mean μ and variance Σ
⊗	Kronecker product
⊙	Hadamard product
$I_{N}$	N×N identity matrix
0	Vector with all zero elements
const	Constant
w.r.t.	With respect to
p(a|b)	Conditional probability density distribution of variable a w.r.t. variable b
p(a;b)	Probability density distribution of variable a w.r.t. parameter b
q(⋅)	Probability density distribution
${\langle \cdot \rangle }_{q(\cdot )}$	Expectation with respect to q(⋅)
diag(⋅)	Transforming matrix/vector into vector/matrix diagonally
i.i.d.	Independent and identically distributed
vec(⋅)	Vectorization function
unvec(⋅)	Matrixing function
⌈⋅⌉	Top integral function
mod(a,b)	Function to find the remainder of a divided by b
$A^{n}$	Matrix A to the power of n
tr(⋅)	Function to obtain the trace of a matrix
$A_{i\cdot }$,$A_{\cdot j}$	The *i*-th row of matrix **A** and the *j*-th column of matrix **A**
$A_{i,j}$	The element in the *i*-th row and *j*-th column of matrix **A**
${\left\|\cdot \right\|}_{p}$	Obtain $l_{p}$ norm for each row of a matrix
${\left\|\cdot \right\|}_{p.q}$	Obtain $l_{p}$ norm after finding $l_{p}$ norm
E(⋅)	Expectation
$1_{L\times L}$	L×L Matrix with all elements 1
|⋅|	Function to find absolute value or determinant
arg(⋅)	Function to find phase

**Table 2 sensors-24-02336-t002:** Complexity of various algorithms.

BSBL	BSBL-APPR	BSBL-GAMP	SBL [21]
$O(M^{2}NL^{3})$	$O(M^{2}N)$	O(ML)	$O(M^{2}N^{2})$
L1-SRACV	IC-SPICE	L1-SVD	HSL
$O(M^{3}N^{3})$	$O(MN^{3})$	$O(M^{3})$	$O(2M^{2}N)$
RVM-DOA	RV-ON-SBL	ON-SBLRVM	SBLMC
O(MNL)	$O(M^{3})$	$O(M^{2}NL)$	$O(M^{3}L)$

## Data Availability

No new data were created.

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
