# Peer review of "Direction-of-Arrival Estimation via Sparse Bayesian Learning Exploiting Hierarchical Priors with Low Complexity"

_sensors, 2024, doi:10.3390/s24072336_

Round 1

Reviewer 1 Report

Comments and Suggestions for Authors
  1. - The introduction provides a broad overview without specifying particular applications of the problem the authors aim to address. Direction of Arrival (DOA) or Direction of Departure (DOD) estimation has been extensively studied across various telecommunication fields, including innovative environments such as vehicular environments and Reconfigurable Intelligent Surfaces (RIS). It would be beneficial to provide references to better contextualize the problem within these domains. For instance, Mizmizi, Marouan, et al. "Fastening the initial access in 5G NR sidelink for 6G V2X networks." Vehicular Communications 33 (2022): 100402

  2. examines spatial synchronization by assuming prior knowledge of DOA distribution in environments, while A RIS-based vehicle DOA estimation method with integrated sensing and communication system." IEEE Transactions on Intelligent Transportation Systems (2023)

  3. investigate a RIS-based vehicle DOA estimation method within an integrated sensing and communication system.

  4. - The derivation presented in the paper is intriguing but may be challenging for readers to follow. It would be helpful for the authors to include remarks or explanations at key points to aid comprehension.

  5. It is essential for the proposed algorithms to adhere to standard procedures. Referring to 3rd Generation Partnership Project (3GPP) approaches for DOA estimation could enhance the consistency and credibility of the proposed methods.

  6. - The presentation of results is commendable, with thorough comparisons to other approaches. However, improving the image quality, such as enhancing font clarity and adjusting linewidths, would enhance readability. Additionally, providing clearer conclusions drawn from each figure would help readers better understand the significance of the results.

Comments on the Quality of English Language

Good english

Reviewer 2 Report

Comments and Suggestions for Authors

This paper propose a novel scheme for DOA estimation. Overall, this paper is well written and I only have a few minor comments. 

1. The literature review is not comprehensive enough and a more adequate introduction to the relevant work is needed.

2. The content in Appendix A should follow the Introduction.

3.The contributions of this paper need to be elaborated in more detail.

Comments on the Quality of English Language

English should be improved.

Reviewer 3 Report

Comments and Suggestions for Authors

This paper develops a Block-sparse SBL method (BSBL) using hierarchical priors to enhance sparsity efficiently. Some issues need to be addressed as follows.

1.     Authors should emphasize the contributions, novelty, and important findings in the abstract. Besides, abbreviations “BSBL-APPR” and “BSBL-GAMP” should be explained.

2.     As this manuscript is submitted to the section “communications”, authors need to introduce the background about this study in wireless communications. Besides, recent high quality works on communications should be introduced, such as Throughput maximization of wireless-powered communication network with mobile access points, IEEE TWC, 2023.

3.     In the introduction, few references have been introduced in detail. Authors need to compare the contributions of relevant reference and this manuscript in detail.

4.     In the introduction, the contributions of this paper are rather simple. Authors need to summarize the contributions and novelties in detail.

5.     The format of formulas and descriptions should be revised, i.e., “where..” after (1) or (2) should be non indented. The punctuations in formulas should be added.

6.     The formulated problem must be clearly provided and analyzed. It seems the proposed method is introduced after the data model.

7.     How does the proposed method advance in the research field compared with existing methods?

8.     Important observations and reasons behind the figures in simulations should be provided in detail. The descriptions for the simulations are simple.

9.     More recent competitive schemes are expected for comparisons in the simulations.   

 10. Important results and findings should be better summarized in conclusion.

Comments on the Quality of English Language

Extensive editing of English language is required for this manuscript.

Reviewer 4 Report

Comments and Suggestions for Authors

Report on the manuscript titled "DOA Estimation via Sparse Bayesian Learning Exploiting Hierarchical Priors with Low Complexity" by Ninghui Li, Xiaokuan Zhang, Fan Lv, Binfeng Zong. 

In this paper the authors study a specific DOA estimator based on sparse Bayesian learning, having hierarchical priors. They propose some novel extensions of BSBL, denoted as BSBL-APPR and BSBL-GAMP. From  the graphs it is clear that the results offer a support in this direction. Although the analysis is viable, the presentation of the paper can be improved:

- the main problem of the paper is the writing style - it is too telegraphically, without any discussion related to new aspects that are introduced;

-the novelty of the paper it is not clear from the present discussion;

-in the Introduction the authors should expand the discussion and present the state of the art in this direction

-the Conclusion section is too short, without any physical discussion related to the results obtained

-the authors don't cite or relate the algorithms to previous works - the novelty of the analysis is not clear

I view of these arguments, I think that the paper should be rewritten almost from scratch. 

Round 2

Reviewer 1 Report

Comments and Suggestions for Authors

The authors addressed all my comments.

Thank you

Comments on the Quality of English Language

The authors addressed all my comments.

Thank you

Reviewer 3 Report

Comments and Suggestions for Authors

Authors have made great efforts on revising this manuscript. Some minor issues should be addressed as follows: The format of references should be revised, for example, “Liu, aoying, et al.” in [1] should be revised as “Liu X et al.”

Comments on the Quality of English Language

Minor editing of English language is required for this manuscript.

Reviewer 4 Report

Comments and Suggestions for Authors

The authors have improved the manuscript in a sufficient manner.